# aMMP-8 Point-of-Care Test (POCT) Identifies Reliably Periodontitis in Patients with Type 2 Diabetes as well as Monitors Treatment Response

**DOI:** 10.3390/diagnostics13132224

**Published:** 2023-06-29

**Authors:** Anna Maria Heikkinen, Taru Tuulia Sokka, Eeva Torppa-Saarinen, Elina Pimiä, Minna Jokinen, Minna Maijala, Iina Rantala, Taina Tervahartiala, Timo Sorsa, Timo Kauppila

**Affiliations:** 1Faculty of Medicine and Health Technology, University of Tampere, 33100 Tampere, Finland; timo.kauppila@fimnet.fi; 2Wellbeing Services County of Pirkanmaa, 33400 Tampere, Finland; taru.sokka@pirha.fi (T.T.S.); eeva.torppa-saarinen@pirha.fi (E.T.-S.); elina.pimia@pirha.fi (E.P.); minna.l.jokinen@pirha.fi (M.J.); minna.maijala@pirha.fi (M.M.); 3Department of Oral and Maxillofacial Diseases, University of Helsinki, 00014 Helsinki, Finland; iina.a.rantala@helsinki.fi (I.R.); taina.tervahartiala@helsinki.fi (T.T.); timo.sorsa@helsinki.fi (T.S.)

**Keywords:** diabetes, periodontitis, aMMP-8, point-of-care test

## Abstract

Background: The link between diabetes and periodontitis is bi-directional: high glucose levels increase the risk of periodontitis and elevated oral fluid aMMP-8 as well as diabetic development while untreated periodontitis worsens glycaemic control. Methods: Type-2 patients (N = 161) underwent an aMMP-8 Point-of-Care Test (POCT) at diabetes clinics. If the test was positive, the patient was sent to an oral health care clinic and oral health examination, health-promoting as well as necessary treatment procedures were carried out. Only 41 patients underwent full clinical evaluations. At the end of the treatment, an aMMP-8 POCT (B) was performed and if the test was positive, the treatment was continued and a new test (C) was performed, aiming for test negativity. The glycated haemoglobin (GHbA1c) test was performed approximately 6 months from the original appointment. Results: GHbA1c concentrations did not decrease during the follow-up. The concentrations of aMMP-8 assessed by POCT, and clinical parameters decreased. Changes in GHbA1c and aMMP-8 levels assessed by POCT during the treatment correlated positively with each other (*p* < 0.01). Conclusion: aMMP-8 POCT proved its reliability, and that its use is beneficial in the diabetes clinic, it enables identifying patients with periodontal findings reliably and guides them directly to an oral health clinic.

## 1. Introduction

Approximately 500,000 Finns suffer from type-2 diabetes (T2D), and the number is growing every year according to the statistics of the Finnish Institute for Health and Welfare. The link between diabetes and periodontitis is bi-directional: high glucose levels increase the risk of periodontitis and complicate periodontitis and its treatment, while untreated periodontitis worsens glycaemic control in diabetes [1,2]. The mechanism of diabetes promoting by periodontitis has been described as the following. Together, the receptor for advanced glycation end (RAGE) products expressed by macrophages and advanced glycation end (AGE) products produced by T2D can activate the release of proinflammatory cytokines [3]. Furthermore, Chee et al., (2013) observed that the increase in high inflammatory phenotype of monocytes and the changes in adhesion, chemotaxis, and phagocytosis of neutrophils would eventually lead to dysbiosis of oral microbiome with accumulation of periodontal pathogens, and thus induce periodontal inflammation [4]. As a conclusion, TD2 can regulate the host’s response to periodontitis by an adipose factor pathway, increasing cytokines and pathways of AGE/RAGE and the receptor activator of NF-κB (RANK)/receptor activator of NF-κB ligand (RANKL) latter with changing bone metabolism toward to alveolar bone loss [5,6]. Severe periodontitis concerns 10% to 15% of the adult population globally [7]. Periodontitis is associated with an increased risk of several chronic systemic diseases through systemic, often low-grade, inflammation. Such diseases are diabetes [7], inflammatory bowel disease [8] cancers, and cardiovascular diseases [9,10,11,12].

Activated matrix metalloproteinase-8 (aMMP-8) is a major destructive collagenase in periodontitis and a potential biological marker in both incipient periodontitis and advanced periodontitis [13]. To detect the progression of periodontitis and the associated active tissue destruction, the activated matrix metalloproteinase-8 (aMMP-8) Point-of-Care Test (POCT) can be used with a cut-off of 20 ng/mL [14,15,16,17,18]. The practical advantages of this test are its simplicity, speed, cheapness, non-invasiveness, and safety, and the fact that it seems to be more accurate than traditional methods. The study focuses on the Finnish adult population that has been diagnosed with T2D or its precursor (prediabetes) [19]. Prediabetes was diagnosed according to WHO criteria [20]. In the progression of periodontitis, the concentration and activation of aMMP-8 in oral fluids is significantly increased, which can be established using the aMMP-8 POCT. The test has already been used promisingly to support the diagnosis of periodontitis and to identify pre-diabetes patients in the Greek dataset [21]. MMP-8 can fragment and inactivate insulin receptor promoting the diabetic disease development, and synthetic MMP-8 inhibitors, such as doxycycline and batimastat, can prevent this [22]. It is important to know for the future prevention and control of diabetes if periodontitis plays a role in the development of diabetes and its potentially fatal complications [23].

The aim of the study is to clarify putative usefulness of aMMP-8 POCT to support the diagnosis and treatment of incipient periodontitis and periodontitis in treating primary health care patients with T2D. The aim is to be able to identify patients prone to periodontitis and in need of periodontal care as early as possible at the diabetes nurse’s office and to bring them within the scope of targeted preventive and corrective care in oral health care. Furthermore, is the aMMP-8 POCT suitable for use in a primary care operating environment as part of the treatment path for diabetes patients and is it possible to increase cooperation between oral health and general practitioners and support integration and comprehensive patient care with new ways of working.

## 2. Materials and Methods

The implementation is an open progressive pilot study. The data used in the study consisted of Finnish-speaking primary health care patients aged 18 years or older, preliminary data were also collected (age, data on brushing and cleaning interdental spaces, use of tobacco products, Body Mass Index (BMI), glycated haemoglobin (GbA1c), and clinical data describing oral health (Bleeding on Probing (BOP), number of probing pocket depth (PPD) at least 4 mm, decayed tooth (DT), and decay-missing-filled (DMF) index), and result of the aMMP-8 POCT). PPD values were recorded, such as probing pocket depth 4 mm (PPD4), probing pocket depth 5 mm (PPD5) and probing pocket depth ≥ 6 mm (PPD deep). We also recorded PPD total meaning the total amount of ≥4 mm probing deep pockets. Research patients were recruited from the appointments of nurses implementing diabetes guidance at the Linnainmaa Health Clinic of the City of Tampere Diabetes Outpatient Clinic Diabetes Clinic in Finland. The primary health care services of the City of Tampere follow regional and national treatment recommendations and treatment paths in the treatment of diabetics. Diabetes patients underwent an aMMP-8 Point-of-Care Test (POCT) at during their scheduled diabetes visits. If the test (A) was positive (two lines), the patient was sent to an oral health care appointment, where an oral health examination was carried out using BOP, PPD ≥ 4 mm, DT, and DMF the necessary treatment was carried out in accordance. At the end of the treatment, an aMMP-8 POCT (B) was performed, and if the test was positive the treatment was continued and a new test (C) was performed. An effort was made to obtain a test for negativity (only one line). The GHbA1c test was performed 6 months (±2 months) from the original appointment with the public health nurse. The aim was to have a total of 200 patients in the study, but due to the COVID-19 pandemic, this was not possible and a total of 51 patients participated in the study. Data are presented as mean (95%) confidence interval (CI)), or raw values (in Figure 1C–E). Medians with 25 and 75% quartiles are presented when appropriate. The research permit was obtained from the City of Tampere and the ethical statement of Karolinska Institute (2016-08-24/2016/1:8 and 2016-1-24 Ref: 2016/1410-31/1).

Magnitude differences in GHbA1c, aMMP-8 POCT, and clinical parameters were analysed as comparisons between before–after situations paired with a *t*-test or Wilcoxon signed rank-test, when applicable. Correlations between GHbA1c, aMMP-8 POCT, and clinical parameters were determined with Pearson or Spearman correlation, when applicable. *p* < 0.05 was considered as a statistically significant difference.

## 3. Results

There were 51 patients who underwent the two biochemical analyses and 41 patients who underwent full clinical evaluation with two examinations. Twenty-nine of them (57%) were men. The mean age of the patients was 65.8 years. Thirty-one had T2D diagnose and 20 had prediabetic glucose levels. The mean BMI was 33.5 (1.8; CI 95%). The mean duration of oral treatment was 9.5 (1.5; CI 95%) months.

GHbA1C concentrations did not decrease during the follow-up (Table 1).

The concentrations of aMMP-8 POCT decreased (Figure 1A). The percentage of BOP (Figure 1B), number of PPD deep (Figure 1C), PPD total (Figure 1D), and DT decreased (Figure 1E) too. Moreover, numbers of PPD4 and PPD5 decreased during the treatment (Table 1). Values of DMF were not statistically significantly altered (Table 1).

Before treatment, GHbA1C and aMMP-8 POCT concentrations did not correlate with each other (r = 0.209). In this stage, aMMP-8 POCT concentrations correlated positively with numbers of PPD5, PPD total, and percentages of BOP. GHbA1C concentrations correlated positively with DT (Table 2). After treatment, no correlations were found between GHbA1C or aMMP-8 POCT concentrations and clinical parameters were not statistically significant. There were no statistical differences in the concentrations of aMMP-8 between groups in brushing their teeth and interdental spaces, or consuming tobacco products.

Changes in GHbA1C and aMMP-8 POCT concentrations during the treatment correlated positively with each other (*p* < 0.01, Figure 2). Neither of them correlated with any of the changes in clinical parameters.

## 4. Discussion

The key findings of this study were that periodontal findings can be conveniently identified in real time and reliably in diabetics, furthermore, the identified diabetics benefited from the periodontal treatment. This was reflected in improved periodontal clinical indices (PPD and BOP values) as well as with reduced aMMP-8 test results. Importantly, the decrease in GHbA1c correlated positively with the decrease in the aMMP-8 test, the GHbA1C values decreased more as the test results decreased. Cochrane Database of Systematic Review of Simpson et al., (2022) [24] stated that evidence from 30 studies showed an absolute reduction in GHbA1C of 0.43% (4.7 mmol/mol) 3 to 4 months after treatment of periodontitis. Furthermore, they observed that after 6 months (from 12 studies) an absolute reduction in GbA1C was 0.30% (3.3 mmol/mol), and after 12 months (from one study) it was 0.50% (5.4 mmol/mol), respectively. Baeza et al., (2020) [25] reported in their meta-analyses that conventional periodontal treatment can improve metabolic control and reduce systemic inflammation in patients with T2D by reducing serum levels of GbA1C as well as C-reactive protein (CRP). On the other hand, we could not directly demonstrate that periodontal therapy would have been able to improve diabetes control, but almost half of our patients had normal GHbA1c values and therefore no major GHbA1c-changes were possible in this pre-diabetic population. Yet, the present result is in line with recent randomized controlled trial (RCT) findings suggesting that oral health care improves oral health-related quality of life in primary health care patients with T2D [26]. There are many factors that are affecting the balance of diabetes care, and oral health is one among those.

Another important finding was that the aMMP-8 POCT worked well for monitoring periodontal treatments in this diabetic patient group. The test differentiated patients with deep pockets from those who had healthy gingival status. Thus, it did not suggest that healthy patients were diseased. Testing aMMP-8 can target care to those who need it and thus can reduce over care. This is in line with the study by Heikkinen et al. (2016) [15] which reported that among adolescents the test had specificity of 100%; thus, no fake positives [15]. In general, previous studies have observed that sensitivity and specificity of oral fluid aMMP-8 POCT (with a cut-off of 20 ng/mL) for periodontitis and peri-implantitis have been found to be 76–90% and 85–96%, respectively, depending on the clinical definitions of the periodontal diseases [27,28,29]. Recently, aMMP-8 POCT has been reported to have an applicability in real-time quantitative diagnostics for both diabetes and periodontitis at the dentist’s office [21,30]. It is noted that “if chairside methods of HbA1c assessment are unavailable, the combination of periodontitis, increasing age, BMI, and aMMP-8 appears to be a viable screening strategy for correctly referring dental patients to their physicians for further prediabetes/and diabetes testing” [31]. In fact, diabetes and diabetic development up-regulate and activate MMP-8 in gingival tissues and oral fluids [32,33,34], and MMP-8 and MMP-9 inhibitor, doxycycline, can inhibit this [22,32,33]. In this study, correlation between GHbA1c and aMMP-8 levels assessed by aMMP-8 POCT provide further evidence of the previous studies [32,33,34].

The strength of this study is that patients were monitored and treated in an oral health clinic for a long time, almost 12 months. They were treated according to national periodontal guidelines recommendations as they would otherwise be treated. The weakness of the study was, compared with expectations, a quite small data amount due to the cancellation of non-urgent treatments due to COVID-19. This also applied to patients with diabetes whose treatment times had to be postponed further, which increased the number of dropouts. Another weakness was the lack of a control group. In the study of Kumar et al., (2006) [35] they observed that higher concentrations of MMP-8 as well as MMP-9 were reported in the gingival tissues and oral fluids of diabetic patients with periodontitis. MMP-8, matrix metalloproteinase-9 (MMP-9), and matrix metalloproteinase-13 (MMP-13) in gingival crevicular fluid (GCF) are suggested to be related to metabolic syndrome and periodontitis [36]. Additionally, Miller et al., (2021) reported that salivary MMP-8 and IL-1β can discriminate periodontitis in T2D patients in their study with 92 participants [37]. This in line with Hardy et al., (2013), as they noted an increasing trend in MMP-8 protein expression levels in periodontal tissues from patients with both periodontal disease and diabetes [38]. Assessing total MMP-8 and MMP-9 in oral fluids will eventually reduce the diagnostic utilization and exactness [30,31] of these biomarkers in comparison with aMMP-8 POCT, targeting activated MMP-8, which is suitable as shown in this study and previous online and real-time monitoring studies as well as in diagnostics [21,32,39]. This study is more of an initial study, and in the future, there will be a need for RCT studies on this topic, as well as more studies related to other biomarkers such as active MMP-9 and its regulators. 

## 5. Conclusions

The aMMP-8 test proved its reliability, and that its use is beneficial in the diabetes clinic, it allows to reliably identify patients with periodontal findings as well as monitor response to treatment. The periodontal health of diabetics is improved with proper treatment and can be demonstrated both by clinical parameters and by the aMMP-8 POCT.

## Figures and Tables

**Figure 1 diagnostics-13-02224-f001:**
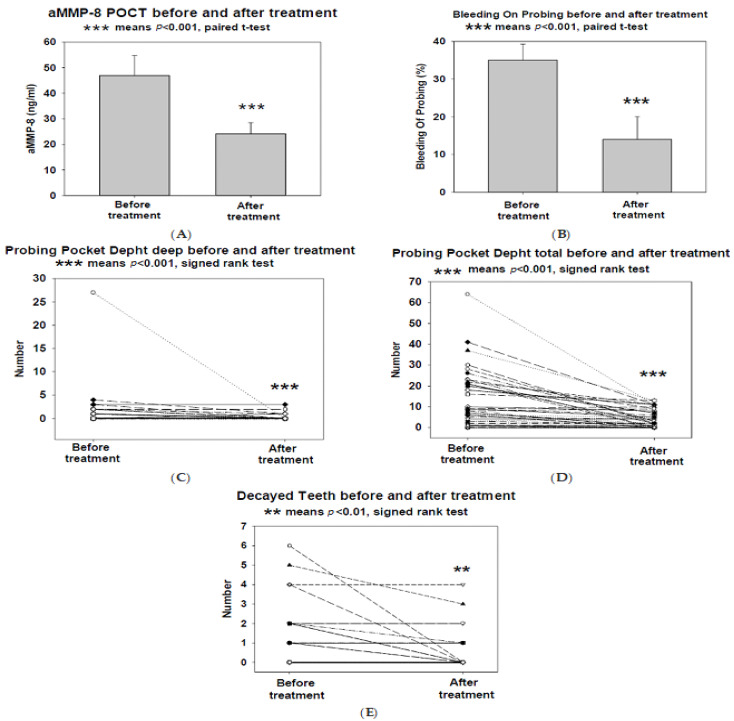
The concentrations of aMMP-8 Point-of-Care Test (POCT) (**A**) and percentages of bleeding on probing BOP (**B**) before and after treatment. Bars show the means, and the brackets show CI 95%. Numbers of PPD deep (probing pocket depth ≥ 6 mm) (**C**), PPD (probing pocket depth) total (**D**), and the number of decayed tooth (DT) (**E**) before and after treatment. Symbols (dots) express values of individual patients before and after treatment.

**Figure 2 diagnostics-13-02224-f002:**
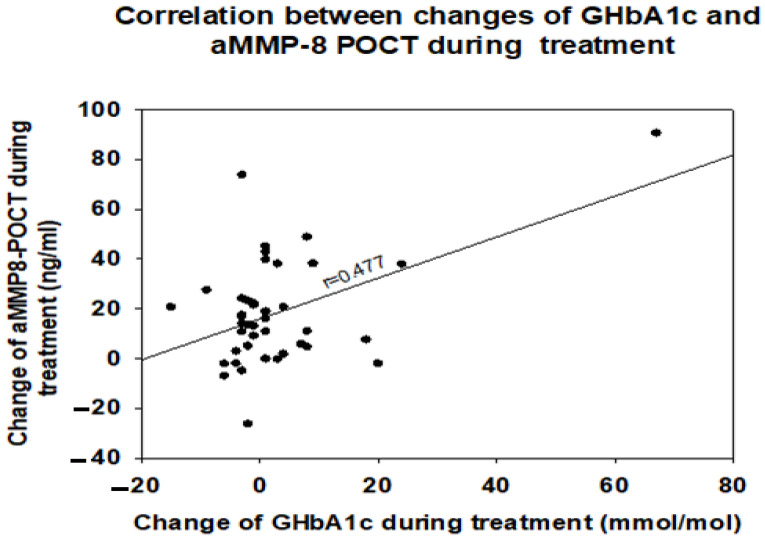
Correlation of changes in GHbA1c and aMMP-8 POCT concentrations during the treatment. Dots express changes in GHbA1c and aMMP8-POCT concentrations in individual patients. Correlation coefficient (r) is shown.

**Table 1 diagnostics-13-02224-t001:** Upper part shows systemic and periodontal indices with test results before and after treatment. Means (with (CI 95%)) or medians (with (25 and 75% quartiles)) are shown when appropriate. *** stands for *p* < 0.001 (Wilcoxon matched pair or paired *t*-test when appropriate).

Systemic and Periodontal Indices with Test Results before and after Treatment.		
Parameter	Before Treatment	After Treatment
Glycated haemoglobin (GHbA1c) (mmol/mol)	52.9 [5.1]	49.9 [3.4]
Number of probing pocket depth 4 mm (PPD4)	6.0 [1.0 to15.5]	2.0 [0.0 to 6.0] ***
Number of probing pocket depth 5 mm (PPD5)	1.0 [0.0 to 2.0]	0.0 [0.0 to 1.0] ***
Decay-missing-filled (DMF) index	23.0 [19.5 to 26.0]	23.0 [19.5 to 26.0]

**Table 2 diagnostics-13-02224-t002:** Presents correlations between clinical parameters and aMMP-8 POCT or GHbA1c concentration in the beginning of the study. Correlation coefficient (r) is shown. * Stands for *p* < 0.05.

Correlations between Clinical and Main Biochemical Parameters		
Clinical Parameter	aMMP-8 Point-of-Care Test (POCT) Concentration	Glycated Haemoglobin (GHbA1c) Concentration
decayed teeth (DT)	*r* = 0.161	*r* = 0.324 *
decayed-missing-filled (DMT) index	*r* = 0.081	*r* = 0.094
probing pocket depth 5 mm (PPD5)	*r* = 0.308 *	*r* = 0.127
probing pocket depth ≥ 6mm (PPD deep)	*r* = 0.037	*r* = 0.229
probing pocket depth total number (PPD total)	*r* = 0.344 *	*r* = 0.079
bleeding of probing (BOP)	*r* = 0.298 *	*r* = −0.101

## Data Availability

The data presented in this study are available on request from the corresponding author.

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
