# Peer review of "aMMP-8 Point-of-Care Test (POCT) Identifies Reliably Periodontitis in Patients with Type 2 Diabetes as well as Monitors Treatment Response"

_diagnostics, 2023, doi:10.3390/diagnostics13132224_

Round 1
Reviewer 1 Report
Explain în introduction to the mechanism of action of diabetes on the periodontal tissues.
Systematization of the results in a table that includes periodontal indices, relevant systemic indices and test results before and after treatment.
Author Response
First, I would like to thank you for your valuable comments. Please see attached our response, the corrections have been made and highlighted in yellow in the main manuscript.
Point 1: Explain in introduction to the mechanism of action of diabetes on the periodontal tissues.
Response: Thank you, were have added more of this issue, please see page 1, lines 33-44.
Point 2: Systematization of the results in a table that includes periodontal indices, relevant systemic indices, and test results before and after treatment.
Response: The Table 1a and 1b are added as suggested.
Furthermore, reference list has been corrected taken account new added references. Few minor corrections have been made also. Two authors, who were inadvertently omitted have now been added. All changes are highlighted with yellow.
Reviewer 2 Report
This is a very interesting article regarding the link between diabetes and periodontitis : high glucose levels increase the risk of periodontitis and elevate oral fluid aMMP-8 as well as diabetic development while untreated periodontitis worsens glycaemic control.
The aim of this atudy is very interesting and actual.
From figure 1, part E is not alligned.
Please provide more recent published articles in the discussion section.
I suggest just moderate english edit.
Author Response
First, I would like to thank you for your valuable comments. Please see attached our response, the corrections have been made and highlighted in yellow in the main manuscript.
Point 1: From figure 1, part E is not aligned.
Response: Figure 1A-E with art E is now aligned, please see corrected Figure1A-E.
Point 2: Please provide more recent published articles in the discussion section.
Response: Thank you, we have added text with more recent published articles in the discussion section, please see page 4, lines 252-258.
Furthermore, reference list has been corrected taken account new added references. Few minor corrections have been made also. English has been moderated slightly. Two authors, who were inadvertently omitted have now been added. All changes are highlighted.